# Second-Line Treatment of Metastatic Renal Cell Carcinoma in the Era of Predictive Biomarkers

**DOI:** 10.3390/diagnostics13142430

**Published:** 2023-07-20

**Authors:** Andreea Ioana Parosanu, Catalin Baston, Ioana Miruna Stanciu, Cristina Florina Parlog, Cornelia Nitipir

**Affiliations:** 1Department of Medical Oncology, Elias Emergency University Hospital, 011461 Bucharest, Romania; andreea-ioana.parosanu@drd.umfcd.ro (A.I.P.); ioana-miruna.stanciu@drd.umfcd.ro (I.M.S.); cristina-florina.pirlog@drd.umfcd.ro (C.F.P.); cornelia.nitipir@umfcd.ro (C.N.); 2Department of Oncology, Faculty of Medicine, “Carol Davila” University of Medicine and Pharmacy, 050474 Bucharest, Romania; 3Department of Urology, Fundeni Clinical Institute, 022328 Bucharest, Romania; 4Department of Urology, Faculty of Medicine, “Carol Davila” University of Medicine and Pharmacy, 050474 Bucharest, Romania

**Keywords:** mRCCs, late-line treatment, prognostic factors, risk classification

## Abstract

Background: Over the past few years, significant advancements have been achieved in the front-line treatment of metastatic renal cell carcinomas (mRCCs). However, most patients will eventually encounter disease progression during this front-line treatment and require further therapeutic options. While treatment choices for mRCCs patients are determined by established risk classification models, knowledge of prognostic factors in subsequent line therapy is essential in patient care. Methods: In this retrospective, single-center study, patients diagnosed with mRCCs who experienced progression after first-line therapy were enrolled. Fifteen factors were analyzed for their prognostic impact on survival using the Kaplan–Meier method and the Cox proportional hazards model. Results: Poor International Metastatic RCCs Database Consortium (IMDC) and Memorial Sloan-Kettering Cancer Center (MSKCC) risk scores, NLR value > 3, clinical benefit < 3 months from a therapeutic line, and the presence of sarcomatoid differentiation were found to be poor independent prognostic factors for shortened overall survival. Conclusions: This study provided new insights into the identification of potential prognostic parameters for late-line treatment in mRCCs. The results indicated that good IMDC and MSKCC prognostic scores are effective in second-line therapy. Moreover, patients with NLR < 3, no sarcomatoid differentiation, and clinical benefit > 3 months experienced significantly longer overall survival.

## 1. Introduction

Kidney cancer is not just one disease but one with great complexity. Renal cell carcinomas (RCCs) are a heterogeneous group of disorders with unique distinct morphological and biological profiles. The 5th edition WHO Classification of Urinary and Male Genital Tumors now recognizes more than seventeen RCC subtypes. Clear cell renal cell carcinoma (CCRCC) is the most common histological subtype of kidney cancer, which comprises 65–70% of all RCCs [1]. 

RCCs rank as the 14th most frequently diagnosed cancers worldwide with approximately 430,000 new cases and nearly 180,000 associated deaths estimated in 2020. Its incidence and mortality trends have changed over time [2]. Over the past two decades, the incidence of RCCs increased by 1–2% annually. In contrast to the rising incidence rates, the overall mortality rate from RCCs has decreased from 69.71% in the early 1990s to 38.96% in the late 2010s. This improvement can be attributed to significant advancements in treatment [3,4,5].

Currently, approximately one-third of RCCs cases are diagnosed with metastatic disease and a poor five-year survival rate of 17% [6,7]. Unfortunately, patients diagnosed with advanced unresectable or metastatic RCCs often experience disease progression during front-line treatment, and only 60% of them survive long enough to receive second-line therapy. As a result, there has been a recent introduction of numerous second-line treatment regimens for mRCC with promising results (Figure 1) [8]. 

The treatment landscape is rapidly evolving, and there are currently several therapeutic options available for second-line therapy, including dual immune checkpoint inhibition (ICI) or a combination of immunotherapy ICIs and antiangiogenic tyrosine kinase inhibitors (TKI) [9].

The second-line therapeutic approaches are determined by the mechanisms of acquired resistance to first-line regimens and require alternative treatment options.

For patients who experience progression on immunotherapy as their first-line treatment, a TKI inhibitor should be considered a second-line therapy option. Conversely, if an ICI plus TKI combination was utilized as the initial therapy, patients should receive a regimen that includes a different TKI either in monotherapy or in combination with immunotherapeutic agents [10,11,12].

Despite recent advancements in mRCC, there is limited knowledge regarding prognostic parameters for second-line therapy. Several prognostic models have been refined over time [13,14,15,16,17]. One of the most widely used models is the Motzer score or Memorial Sloan Kettering Cancer Center (MSKCC) risk model, initially developed in the cytokine era and validated in molecular-targeted therapy [18,19]. Similarly, the International Metastatic Renal Cell Carcinoma Database Consortium (IMDC) model, proposed by Heng et al., was developed and validated for patients receiving first-line antivascular endothelial growth factor therapy. Most recently, the IMDC model demonstrated prognostic value in patients receiving immune checkpoint inhibitors [20,21,22]. 

The IMDC and MSKCC models for prognosis in mRCCs and the front-line therapy administered have been identified as the only prognostic factors in patients receiving second-line therapy [23,24,25]. Previous reports indicated that patients with mRCC and early progression on the front-line regimen had a significantly higher probability of not receiving subsequent therapy. Moreover, there is good evidence that the timing of treatment benefits may serve as a reliable surrogate endpoint for progression-free survival [26,27].

There are several reasons why inflammation is a recognized hallmark of cancer (Figure 2) [28]. First, increased neutrophils can lead to excessive inflammation and a pro-inflammatory condition. Second, neutrophils can reduce lymphocyte T proliferation and inhibit antitumor T-cell responses. Lastly, lymphocyte depletion reflects a defective host immune system that fails to control tumor occurrence and growth.

In the context of cancer, a neutrophil-to-lymphocyte ratio (NLR) indicates an optimal balance between pro- and antitumor immunity [29,30]. NLR is a well-known prognostic marker in several malignancies, including RCCs. Elevated pre-treatment NLR was associated with poor survival and an unfavorable response to first or subsequent-line therapy treatment [31,32].

A particular feature of RCCs is sarcomatoid differentiation, which indicates aggressive behavior and poor prognosis. Even a small amount of sarcomatoid differentiation might independently predict poor overall survival and limited therapeutic options. Also, when adjusted for therapeutic response, patients with sarcomatoid RCCs have worse survival [33,34]. 

However, additional research warrants validation of these potential markers’ prognostic accuracy and consistency in mRCCs, especially when treated with late-line therapy. To address this knowledge gap, we conducted the following retrospective analysis to evaluate the prognostic significance of clinical and biological factors in patients undergoing second-line therapy.

## 2. Materials and Methods

This is a single-center retrospective observational study of mRCCs patients treated in our oncology department. Clinical data were extracted from patients’ medical records in concordance with the recommendations of the ethics committee and the Declaration of Helsinki.

The principal inclusion criteria for the study were documented metastatic clear cell renal cell carcinomas (mCCRCCs), progression on first-line therapy, available clinical and imaging data before initiation of each treatment line, willingness to provide written informed consent, and age ≥ 18 years.

Exclusion criteria include short-term follow-up (<6 months), active autoimmune disease, evidence of active infection before initiating any systemic therapy, second primary cancers, brain metastases, and histotypes other than clear cell renal cell carcinoma.

Between January 2020 to October 2022, 74 patients diagnosed with mCCRCCs initiated first-line therapy. The majority of patients (89.4%) had received first-line treatment with TKI (57.3% sunitinib, 41.1% pazopanib, 1.4% sorafenib), while only 10.5% had received immunotherapy with nivolumab and ipilimumab. During the follow-up period, out of 74 patients treated in our department, 51.3% (38 patients) required second-line treatment. Nivolumab (3 mg per kilogram) plus ipilimumab (1 mg per kilogram) every 3 weeks for four cycles, followed by nivolumab monotherapy, was the most frequently (39.4%) administered second-line regimen, followed by cabozantinib (26.3%) at a daily dose of 60 mg orally once daily, axitinib (21%) at a dose of 5 mg orally twice daily, and pazopanib (13.1%) at a dose of 800 mg orally twice daily. 

Patients were followed from the date of first-line administration to the date of last follow-up or death. The median follow-up was 15.3 (interquartile range: 8.3–22.6) months. The median time to progression on first-line therapy was 7.5 months (HR = 0.49; 95% CI: 0.215–1.65). As of the last follow-up in October 2022, thirty-three patients receiving second-line therapy were alive, while five died. The median OS was 10.5 (range 5–22.5) months. Treatment and therapeutic monitoring were conducted based on computed tomographic scans performed every 3 months.

The data were analyzed using SPSS 23.0. The chi-square test and the t-test were used to compare the categorical and continuous variables, respectively. We identified clinical characteristics with *p*-values ≤ 0.1 in the univariate analysis and further included them in the multivariate Cox regression analysis (*p* < 0.05). We further calculated the curves of overall survival (OS) using Kaplan–Meier survival curves and the log-rank test. The optimal cut-off point of the neutrophil-to-lymphocyte ratio (NLR) was determined by the receiver operating characteristic (ROC) curve analysis. OS was defined from the time of first-line treatment initiation until death or the last follow-up. 

## 3. Results

### 3.1. Baseline Patient Characteristics and First-Line Therapy

A total of 74 mCCRCCs patients were included in the study. The median age was 62.8 years (interquartile ranger, 43–88 years), and 70.3% were males. Most patients (77%) had good performance status, defined as a Karnofsky performance status equal to or greater than 80%. 

Before the start of first-line treatment, 62 (85%) patients with clinically diagnosed mCCRCCs underwent surgical intervention (radical or partial nephrectomy), and only 12 (15%) patients were diagnosed based on the results of a tissue biopsy of the primary tumor. The most frequently observed metastatic sites were the lymph nodes (13.5%), lung (23%), liver (39.2%), and bones (28.4%). The most common WHO/ISUP (WHO/International Society of Urological Pathology (ISUP) grading system) grades were II (43.2%) and III (44.6%). Only 12.2% of tumors were WHO/ISUP grade IV. At Histopathological analysis, sarcomatoid differentiation was evident in seven (9.5%) cases.

The baseline characteristics are presented in Table 1.

We estimated the prognostic scores using baseline clinical and laboratory characteristics. Over 50% of patients with mCCRCCs at first-line treatment were classified as intermediate-risk, according to the IMDC (51.4%) and MSKCC (66.2%) prognostic models.

In the first-line therapy, 89.4% of patients received tyrosine kinase inhibitors (57.3% sunitinib, 41.1% pazopanib, 1.4% sorafenib), and only 10.6% of patients received immunotherapy with nivolumab plus ipilimumab. The median PFS was 7.5 months in patients treated with first-line therapy (Figure 3).

### 3.2. Patient Characteristics before the Start of Second-Line Systemic Treatment

A total of 38 (51.3%) patients were considered resistant to first-line treatment. The mean age was 63.9 years; 71.1% of the patients were males, and 48.7% had a good performance status prior to second-line initiation. Radical (65.7%) and partial nephrectomy (26.3%) were the most common surgical treatments, and only 7.8% of patients underwent tumor biopsy. Two (7.8%) patients had sarcomatoid features in their tumors.

Table 2 summarizes patients’ demographics and characteristics.

The prognostic risk scores were calculated at the start of the second-line therapy with most patients classified as having intermediate risk according to both the IMDC (68.4%) and MSKCC (81.6%) risk classifications. In second-line therapy, 15 patients (39.5%) received immunotherapy with Nivolumab plus Ipilimumab, 10 (26.3%) with Cabozantinib, 5 (13.1%) with Axitinib, and 8 (21.2%) with Pazopanib.

We calculated the neutrophil-to-lymphocyte ratio in all patients receiving second-line therapy. The median NLR value was 2.85 ± 2.05 with a range between 0.8–10. We identified the cut-off point using ROC curves in logistic regression (Figure 4). We obtained a value of three as the optimal cut-off point, and we further divided patients into high (>3) and low (>3) NLR.

### 3.3. Statistical Analysis

We carried out exploratory investigations of the parameters associated with better outcomes in mCCRCCs patients treated with second-line therapy. In particular, it was queried whether different clinical, biological, or histological factors might be associated with higher OS. 

Univariate analysis (Table 3) demonstrated that the high IMDC (*p* = 0.028) and MSKCC (*p* = 0.003) scores, as well as elevated NLR (*p* = 0.002), were significant prognostic factors for poor OS.

There were no statistical differences in terms of gender (*p* = 0.13), tumor location (*p* = 0.26), time from diagnosis to initial systemic treatment (*p* = 0.29), performance status (*p* = 0.33), LDH (*p* = 0.30) hemoglobin (*p* = 0.53), calcium (*p* = 0.25), platelets (*p* = 0.50) or neutrophils (*p* = 0.31).

The multivariate analysis offered a more complete examination of these data. NLR was the only independent predictor of OS with an HR = 8.672 and *p* = 0.010 (Table 4; Figure 5).

The Kaplan–Meier survival curves of NLR showed significant differences in OS between patients with low (<3) and high (>3) NLR.

Time without disease progression was essential for all patients. The duration of treatment and its impact on prognosis was also examined (Table 5). Patients who experienced a clinical benefit for more than three months on a therapeutic line had better overall survival in the first line (22.25 months versus 7.62 months) and the second line (25.22 months versus 12.33 months).

Most patients had a multimodal treatment consisting of surgery and systemic therapy.

Almost 90% of patients had undergone initial surgical resection, including radical nephrectomy with lymph node dissection (24.3%) or without lymph node dissection (55.4%) and partial nephrectomy (9.5%). Minimally invasive biopsy procedures were performed in 10% of cases.

The overall survival did not differ significantly between patients treated with radical nephrectomy (21.73 ± 18.24 months) and partial nephrectomy (22.43 ± 23.25 months) but was reduced in patients with a diagnostic biopsy (8.88 ± 5.98 months), *p* = 0.150 (Table 6, Figure 6).

Unfortunately, we found no statistically significant correlation between the surgical approach and clinical benefit in first and second-line therapy (Table 7).

Another interesting aspect was the sarcomatoid differentiation, which was observed in two (7.8%) patients with metastatic mCCRCCs receiving second-line therapy (Table 8, Figure 7). The results suggested that sarcomatoid features were associated with worse clinical outcomes (14.86 months OS versus 19.97 months OS, *p* = 0.033).

Moreover, the clinical benefit was significantly correlated with the sarcomatoid differentiation in first therapy (*p* = 0.004) and second-line therapy (*p* = 0.029) (Table 9).

## 4. Discussion

RCCs are highly vascular tumors with strong immunogenicity and an unpredictable natural history. The treatment of advanced and metastatic RCCs has revolutionized in the past three decades. Interferon and interleukin two immunotherapies have been a treatment options for over 30 years. Recently, immune checkpoint inhibitors and vascular endothelial growth factor receptors have become promising therapeutic strategies [35]. 

The TNM classification is the most important prognostic factor in mCCRCCs patients treated with first-line therapy. Other independent predictors of long-term survival include MSKCC and IMDC risk models. Few sources of data on potential prognostic factors in mCCRCCs patients receiving second-line therapy exist. Despite the significant progress in the treatment landscape of metastatic RCCs, only a minority of patients have access to second-line therapies [21]. Therefore, understanding the baseline parameters of patients who are eligible for second-line therapy is crucial. This study provides real-world evidence on the characteristics and outcomes of these particular patients.

In this study, 51.3% of mCCRCCs patients received second-line therapy, which is consistent with previous reports [21,36,37]. We evaluated the baseline characteristics of the patients before initiating second-line therapy. Most of the patients had intermediate-risk levels based on the prognostic models.

The IMDC and MSKCC models were originally developed and validated for metastatic mCCRCCs patients who received front-line treatment [38,39,40]. Recently, these models have been shown to profile risk in late-line settings.

The MSKCC is the most used tool for prognostic stratification of patients with mCCRCCs receiving second-line therapy. Data from the pivotal second-line studies, including the CheckMate 025 trial demonstrating superior efficacy for nivolumab over everolimus, or the phase III study METEOR, which showed that cabozantinib significantly improved OS and PFS compared to everolimus [41,42].

Furthermore, a large study by Dudani et al. demonstrated that the IMDC score continues to risk stratify patients with mCCRCCs treated with second-line immune checkpoint inhibitors [43].

Our results show that both MSKCC and IMDC have important prognostic value. Patients with favorable risks had significantly longer overall survival compared to those with poor or intermediate scores (HR = 8.907, 2.148–36.935, *p* = 0.004 for MSKCC; and HR = 1.826, 1.068–3.122, *p* = 0.028 for IMDC).

The neutrophil-to-lymphocyte ratio (NLR) is a controversial topic. NLR is a marker of systemic inflammatory response with prognostic significance. It reflects a dynamic relationship between neutrophils, an essential part of the innate immune system, and the adaptive immune responses carried out by lymphocytes. A high NLR has been unequivocally associated with adverse prognoses in many cancers. The evidence indicates that pretreatment NLR could predict recurrence, disease progression, and survival outcomes in RCCs patients [44,45,46]. 

However, NLR varies widely between patients. The normal range of NLR in healthy adults varies between 1 and 3. Values higher than three are considered pathological [47,48,49,50]. Therefore, it is essential to find a cut-off value to mark the lower and the upper limit of NLR associated with prognosis.

We also investigated the optimal cut-off value for NLR. Firstly, we determined the median NLR value for the patients initiating second-line therapy, which was 2.85 ± 2.05. The findings align with the literature, where NLR values between 1.7 and 5 have been reported, with an NLR of 3 being frequently used [51,52,53]. Secondly, using ROC analysis, we identified the optimal NLR cut-off value of 3. Furthermore, we classified patients into high (>3) NLR and low (<3) NLR. The study reported a negative association between high NLR and OS. Clearly, an increased NLR > 3 was associated with worse outcomes in both univariate and multivariate analyses (*p* = 0.005). Therefore, we hypothesized that adding NLR to the established risk models might help improve their prognostic accuracy.

Accumulating evidence has established the role of NLR as a biomarker of increased immune activation in different disorders, including infections, autoimmune diseases, cardiovascular diseases, and metabolic syndromes [54,55]. Even if most studies have explored the prognostic value of NLR in cancer patients with an optimal cut-off value above 3, a grey zone of NLR values between two and three may serve as an early warning of endothelial dysfunction, chronic vascular inflammation, and atherosclerosis [56,57,58]. The chronic vascular inflammatory process plays a role in the pathogenesis of atherosclerosis, hypertension, diabetes, and obesity. Prior evidence has shown that NLR is a reliable biomarker to predict cardiovascular risk. In patients with coronary artery disease, an NLR over 2.13 independently predicted myocardial damage [59].

Furthermore, NLR has also been investigated as a marker of disease activity and a predictor of relapse in autoimmune diseases [60,61]. For example, D’Amico et al. demonstrated that an NLR > 1 in patients with multiple sclerosis strongly predicted disease activity and aggressive evolution [62]. In conclusion, even an NLR value lower than three may be an unfavorable prognostic factor for RCCs patients with underlying comorbidities.

Although growing evidence showed that immune checkpoint inhibitors, targeted therapy combinations, and late lines of therapy improve clinical outcomes, whether increased treatment duration improves survival remains unclear. Prolonged treatment time for mCCRCCs has an important positive impact on cancer-specific survival. But despite recent therapeutic advances in mCCRCCs treatments, challenges remain in managing progressive or recurrent disease. About 10 to 25% of mCCRCCs patients experience rapidly progressive disease on first-line therapy [39]. An established and accepted definition of a rapidly progressive mCCRCCs still needs to be found. 

For example, Chang and his colleagues defined rapid disease progression as occurring within one month of systemic therapy in mCCRCCs [63]. Another study on real-world treatment patterns in mCCRCCs patients by Bersanelli described rapid progression as the time from the start of the first-line to the beginning of second-line treatment ≤ 24 weeks [64].

This study also examined whether treatment duration in first or subsequent therapeutic lines is associated with survival.

Chen VJ et al. defined the term “clinical benefit” as receiving a therapeutic line for at least three months. They found no statistically significant difference in clinical benefit between first- and second-line treatment, but their data confirm that receiving therapy for at least three months correlated with a longer overall survival [26]. Our results also suggest that patients with over 3 months of clinical benefit had a statistically higher median overall survival in both first-line (22.25 months versus 7.62 months) and second-line (25.22 months versus 12.33 months) therapy. Unfortunately, we found no statistically significant correlation between the surgical approach and clinical benefit in first and second-line therapy.

Histopathological features can predict prognosis and may help to stratify patients to receive appropriate therapy. The concept of sarcomatoid RCCs was first comprehensively described by Farrow and colleagues in 1968 under the spectrum of renal sarcomas [65]. However, in 2012, the International Society of Urological Pathology (ISUP) grading system for renal cell carcinoma established that sarcomatoid differentiation is an uncommon histological transformation that can occur in most histological subtypes of RCCs, giving them aggressive biological features and poor prognoses [66,67]. Due to the rarity of these tumors and their exclusion from most studies, limited data exist on the current standard of care associated with their treatment [68,69,70,71]. Most clinical evidence is therefore based on case reports or subgroup analyses. 

A systematic review and meta-analysis showed that immune checkpoint inhibitors were associated with remarkable clinical efficacy in patients with sarcomatoid RCCs [72].

However, agents targeting the vascular endothelial growth factor pathway did not report a significant improvement in patient outcomes [73]. Therefore, regardless of the PD-L1 level, the preferred options are immunotherapy-based regimens.

In this current study, we evaluated the pathological features and the subsequent treatment-related outcomes of patients with sarcomatoid RCCs. In total, seven patients (9.5%) had sarcomatoid features. Only two patients with sarcomatoid differentiation received second-line therapy with a modest response rate and poor clinical benefit (14.86 months OS).

Naturally, the single-center retrospective observational analysis has several limitations, including susceptibility to random errors. The main weaknesses of this study are the retrospective nature and the relatively small sample size of patients. There was also a relatively short follow-up period and a lack of external validation. The second-line therapy administered was not well-balanced, preventing a direct comparison of the efficacy of each regimen. Furthermore, the patients did not receive novel drugs approved in mCCRCCs, which have demonstrated survival benefits. However, we propose that the analysis generates hypotheses and may serve as a basis for investigating these potential prognostic factors in larger studies, particularly in later-line treatment settings.

## 5. Conclusions

No consistent reports regarding prognostic factors in second-line therapy for mCCRCCs had been published prior to this study. Therefore, this study contributed valuable new insights to the literature by shedding light on the prognostic role of biomarkers and emphasizing the importance of risk models in mCCRCCs.

The results indicated that second-line therapy is more effective in patients with favorable IMDC and MSKCC prognostic scores, NLR < 3, and no sarcomatoid differentiation. Moreover, patients with clinical benefit > 3 months on a therapeutic line experienced significantly longer overall survival. 

Large-scale research studies are needed to confirm these observations and to increase our understanding of the best therapeutic sequence in mCCRCCs patients.

## Figures and Tables

**Figure 1 diagnostics-13-02430-f001:**
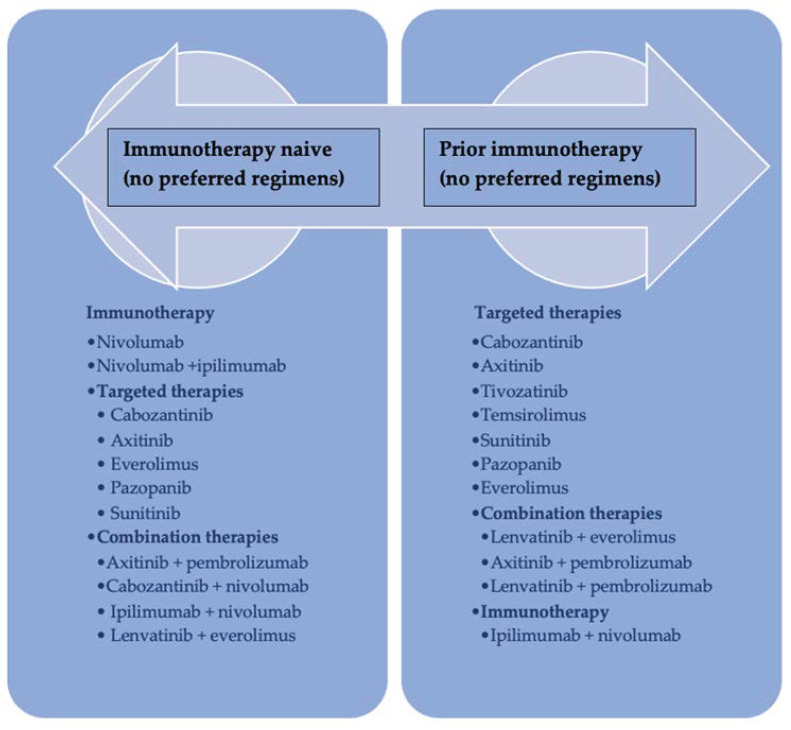
Algorithm for the second-line treatment selection for mRCCs.

**Figure 2 diagnostics-13-02430-f002:**
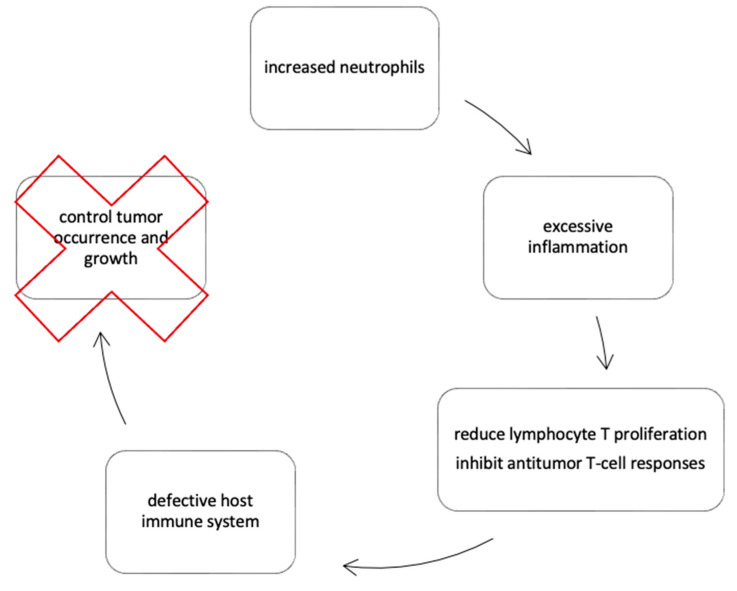
Inflammation as a hallmark and cause of cancer.

**Figure 3 diagnostics-13-02430-f003:**
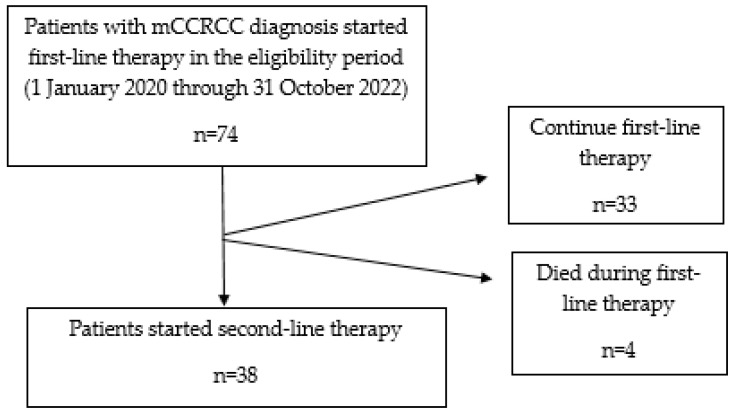
Identification and attrition of the first-line treatment and second-line treatment patient.

**Figure 4 diagnostics-13-02430-f004:**
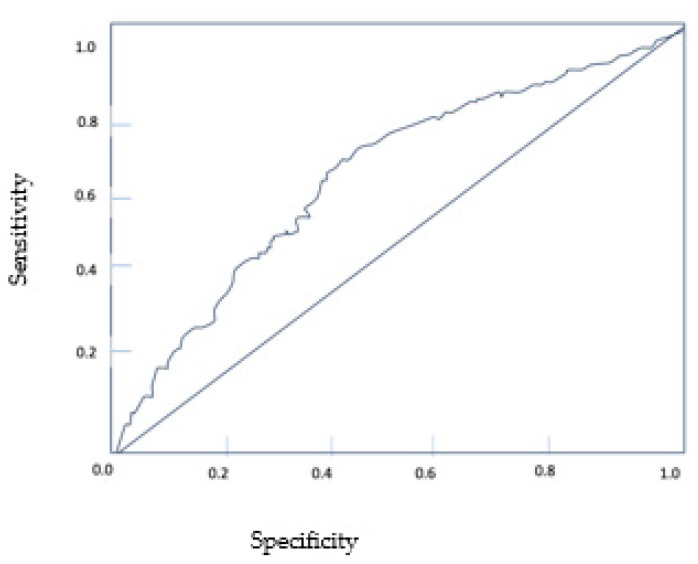
ROC curve for NLR.

**Figure 5 diagnostics-13-02430-f005:**
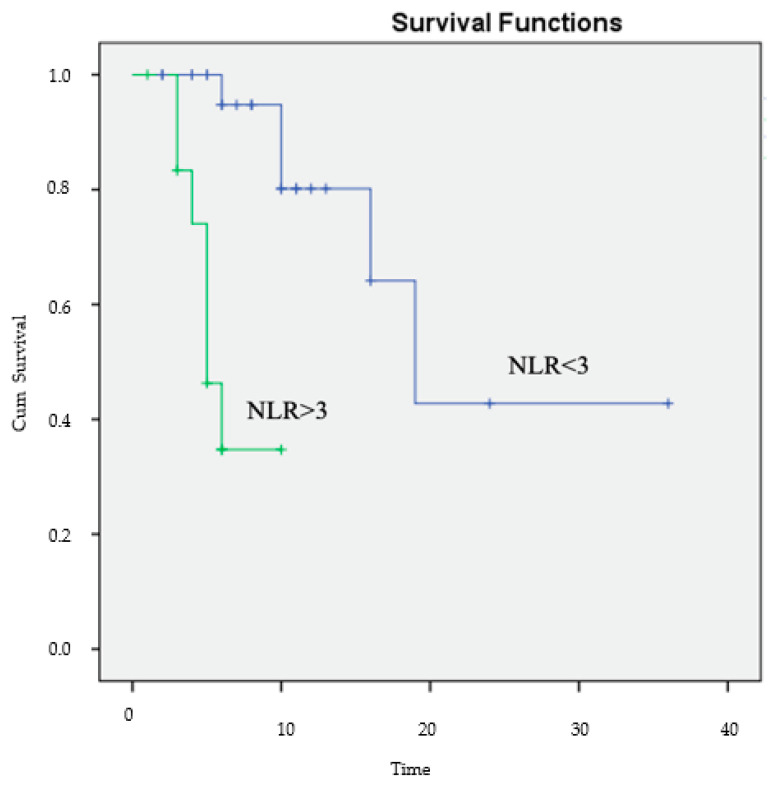
Kaplan–Meier estimates of OS according to the NLR.

**Figure 6 diagnostics-13-02430-f006:**
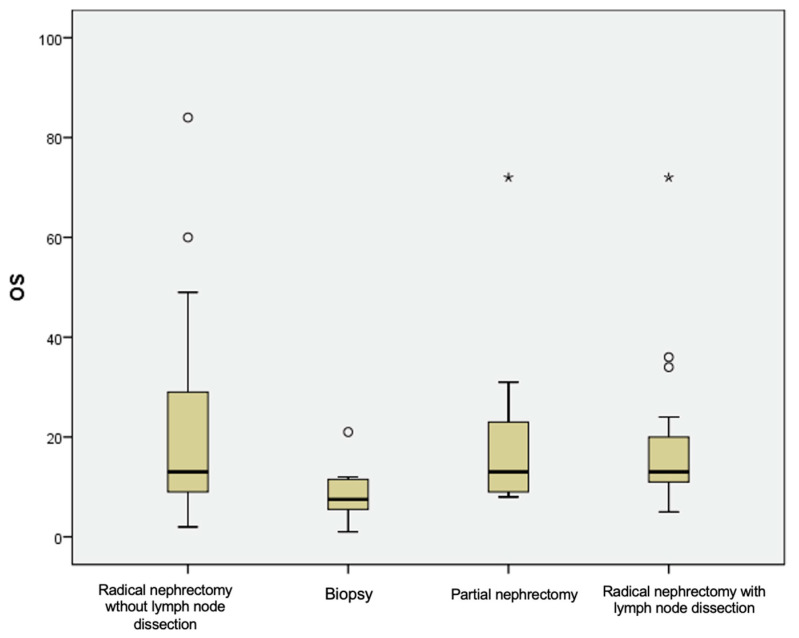
Box plot demonstrating survival difference in mCCRCCs patients according to surgery approach. The outliers (°), the extreme outliers (*).

**Figure 7 diagnostics-13-02430-f007:**
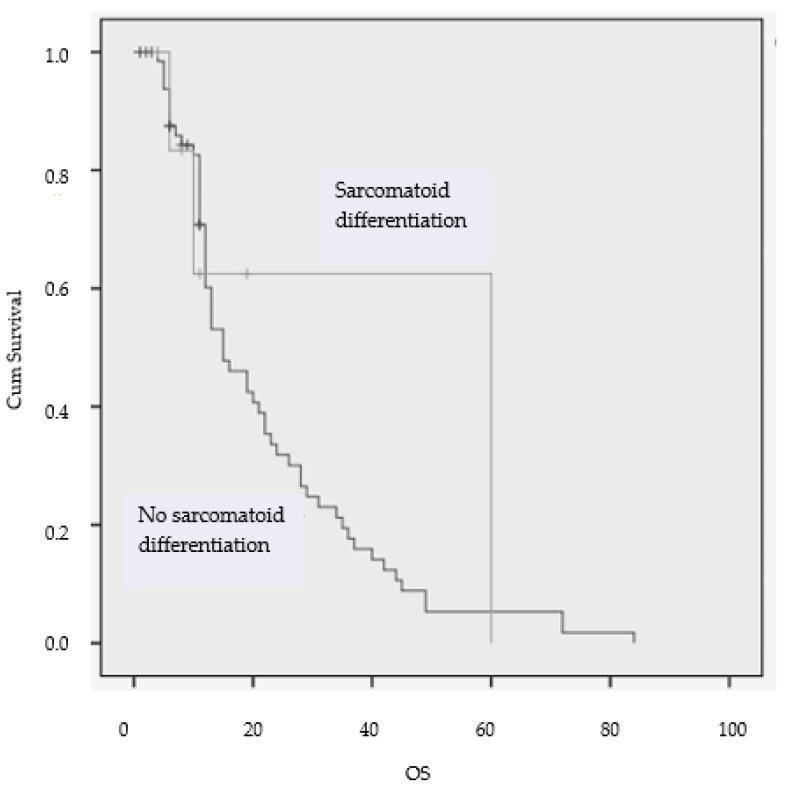
Kaplan–Meier estimates of OS according to the sarcomatoid differentiation.

**Table 1 diagnostics-13-02430-t001:** Demographic and clinical characteristics of patients with mCCRCCs starting first-line therapy.

Patient and Disease Characteristics	Count	Count%
Patients treated with first line therapy	74	
Age median, range (years)	62.8 (range 43–88)	
Gender		
Male	52	
Female	22	
Surgical treatment		
Radical nephrectomy	48	64.8%
Tumor biopsy	11	15%
Partial nephrectomy	15	20.2%
First line therapy		
TKIs	66	89.4%
Immunotherapy	8	10.6%
The main sites of metastasis		
Lung	17	23%
Distant lymph nodes	10	13.5%
Liver	29	39.2%
Bones	21	28.4%
WHO/ISUP grades		
2	32	43.2%
3	33	44.6%
4	9	12.2%
Sarcomatoid differentiation		
Present	67	90.5%
Absent	7	9.5%
Karnofsky Performance Status		
<80%	17	23%
≥80%	57	77%
Time interval from diagnosis to treatment		
<12 months	52	70.3%
≥12 months	22	29.7%
Hemoglobin		
<lower limit of normal	41	55.4%
≥lower limit of normal	43	44.6%
LHD		
≥1.5× upper limit of normal	12	6.2%
<1.5× upper limit of normal	62	83.8%
Serum-corrected calcium		
≥upper limit of normal	16	21.6%
<upper limit of normal	58	78.4%
Platelets		
≥upper limit of normal	11	14.9%
<upper limit of normal	62	83.8%
Neutrophils		
≥upper limit of normal	24	32.4%
<upper limit of normal	50	67.6%
IMDC score		
Favorable	5	6.8%
Intermediate	38	51.4%
Poor	31	41.9%
MSKCC score		
Favorable	8	10.8%
Intermediate	49	66.2%
Poor	17	23%
Poor		

**Table 2 diagnostics-13-02430-t002:** Demographic and clinical characteristics of patients with mCCRCCs starting second-line therapy.

Patient and Disease Characteristics	Count	Count%
Patients treated with second line therapy	38	51.3%
Gender		
Male	27	71.1%
Female	11	28.9%
Age median, range	63.92 ± 8.03 (48–78)	
Karnofsky status		
>80%	18	48.7%
<80%	19	51.3%
Tumor location		
Left side	20	52.6%
Right side	18	47.4%
Sarcomatoid differentiation		
No	35	92.1%
Yes	2	7.8%
Surgical treatment		
Radical nefrectomy	25	65.7%
Partial nefrectomy	10	26.3%
Biopsy	3	7.8%
Second line therapy		
Nivolumab + Ipilimumab	15	39.5%
Cabozantinib	10	26.3%
Axitinib	5	13.1%
Pazopanib	8	21.1%
IMDC score		
Favorable	1	2.6%
Intermediate	26	68.4%
Poor	11	28.9%
MSKCC score		
Favorable	1	10.8%
Intermediate	31	81.6%
Poor	6	15.8%

**Table 3 diagnostics-13-02430-t003:** Univariate analyses for overall survival.

Baseline Parameters	N (%)	Univariate HR (95%CI)	*p* Value
Patients treated with second line	38 (51.3%)		
Gender		0.314 (0.068–1.458)	0.139
Male	27 (71.1%)		
Female	11 (28.9%)		
Age	63.92 ± 8.03	1.048 (0.957–1.148)	0.309
(median, range)	(48–78)		
Tumor location		0.512 (0.157–1.669)	0.267
Left side	20 (52.6%)		
Right side	18 (47.4%)		
Time from diagnosis to initial systemic treatment		2.272 (0.488–10.569)	0.295
<12 months	12 (31.6%)		
>12 months	26 (68.4%)		
Karnofsky status		1.766 (0.554–5.633)	0.336
>80%	18 (48.7%)		
<80%	19 (51.3%)		
Hemoglobin		1.444 (0.454–4.587)	0.534
<12 g/dl	18 (47.4%)		
>12 g/dl	20 (52.6%)		
Platelets		2.030 (0.248–16.615)	0.509
<upper limit of normal	35 (92.1%)		
>upper limit of normal	3 (7.9%)		
LDH		1.443 (0.421–5.224)	0.301
>1.5× upper limit of normal	16 (33.3%)		
<1.5× upper limit of normal	22 (66.6%)		
Corrected calcium		0.421 (0.171–1.655)	0.254
>upper limit of normal	10 (26.3%)		
<upper limit of normal	28 (73.7%)		
Neutrophils		1.261 (0.415–5.503)	0.313
>upper limit of normal	21 (55.2%)		
<upper limit of normal	17 (44.7%)		
NLR (median range)	2.85 ± 2.05 (0.8–10)	1.324 (1.078–1.625)	0.007
NLR		9.599 (2.299–40.072)	0.002
<3	25 (65.8%)		
>3	13 (34.2%)		
IMDC score		1.826 (1.068–3.122)	0.028
favorable	1 (2.6%)		
intermediate	26 (68.4%)		
poor	11 (28.9%)		
MSKCC score		8.907 (2.148–36.935)	0.003
favorable	1 (10.8%)		
intermediate	31 (81.6%)		
poor	6 (15.8%)		

**Table 4 diagnostics-13-02430-t004:** Multivariate analyses for overall survival.

Variables	*p* Value	HR 95.0% CI
NLR	0.010	8.672 (1.658–45.349)
IMDC score	0.320	3.092 (0.334–28.661)
MSKCC score	0.961	506.651 (0.000–8.078)

**Table 5 diagnostics-13-02430-t005:** The therapeutic response and clinical outcome in first and second-line therapy.

Therapeutic Response		Overall Survival				
		Mean	Standard Deviation	Median	Min.	Max.
First line therapy	<3 months	7.62	4.84	8.00	1	19
	>3 months	22.25	18.07	15.00	4	84
Second line therapy	<3 months	12.33	9.58	9.00	6	31
	>3 months	25.22	16.95	19.00	8	72

**Table 6 diagnostics-13-02430-t006:** The surgical approach and clinical outcome.

Surgical Approach	N (%)	OS (Months)
Radical nephrectomy	41 (55.4%)	21.73 ± 18.24
Radical nephrectomy with lymph node dissection	18 (24.3%)	18.72 ± 15.68
Partial nephrectomy	7 (9.5%)	22.43 ± 23.25
Tumor biopsy	8 (10.8%)	8.88 ± 5.98

**Table 7 diagnostics-13-02430-t007:** The therapeutic response, surgical approach and clinical outcome in first and second-line therapy.

Therapeutic Response		Radical Nephrectomy	Tumor Biopsy	Partial Nephrectomy	Radical Nephrectomy with Lymph Node Dissection	*p* Value
		N (%)				
First line therapy	<3 months	9 (21.95%)	1 (12.55)	2 (28.55)	1 (5.55)	0.382
	>3 months	32 (78.05)	7 (87.55)	5 (71.45)	17 (94.45)	
Second line therapy	<3 months	3 (13.65)	1 (25.05)	1 (16.65)	1 (16.65)	0.953
	>3 months	19 (86.35)	3 (75.05)	5 (83.35)	5 (83.35)	

**Table 8 diagnostics-13-02430-t008:** The presence of sarcomatoid differentiation and clinical outcome.

Tumor Caracteristics	N (%)	Overall Survival (Months)
No sarcomatoid differentiation	35 (92.1%)	19.97
With sarcomatoid differentiation	2 (7.8%)	14.86

**Table 9 diagnostics-13-02430-t009:** Clinical benefit for patients with sarcomatoid differentiation.

		Sarcomatoid Differentiation		
	Therapeutic Response	No	Yes	*p* Value
First-line therapy	<3 months	9 (13.43%)	4 (57.14%)	0.004
	>3 months	58 (86.57%)	3 (42.86%)	
Second-line therapy	<3 months	6 (18.18%)	4 (80%)	0.029
	>3 months	27 (81.82%)	1 (20%)	

## Data Availability

Data supporting this study are included within the article and Appendix A.

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
