# Peer review of "Second-Line Treatment of Metastatic Renal Cell Carcinoma in the Era of Predictive Biomarkers"

_diagnostics, 2023, doi:10.3390/diagnostics13142430_

Round 1

Reviewer 1 Report

The author discussed predictive biomarkers about second-line treatment of mRCC. It is not well written, and there are some questions. They need to be well promoted.

1.In the articles and tables, there are a large number of “ , ” should be replaced with "." .Such as Line 109,110,148,154,165,166, Table 2, 4,5,6. 

2. There are many mistakes in the Tables. In Table 3, () is Missing. In Table 6, the % is not inconsistent.  

3. The accuracy of Table 1 should be unified, Line  Tumour biopsy 15.0, Karnofsky Performance Status 23.0% 77.0%, MSKCC score poor 23.0%

Author Response

Authors’ response to reviewer’s 1 comments

We are grateful to the reviewer for the careful evaluation of our manuscript and for the constructive suggestions; all of them helped us to further improve the quality of our manuscript. We appreciate the thorough work of the reviewer and all the comments. We believe that we could fix his suggestions.

Comment 1. In the articles and tables, there are a large number of “ , ” should be replaced with "." .Such as Line 109,110,148,154,165,166, Table 2, 4,5,6. 

Answer: As the reviewer kindly suggested, we have made all necessary changes in our manuscript.

Comment 2.   There are many mistakes in the Tables. In Table 3, () is Missing. In Table 6, the % is not inconsistent.  

Answer: We thank the reviewer for pointing out these errors.  The reviewer is correct, and we have made the changes in the manuscript.

Comment 3. The accuracy of Table 1 should be unified, Line  Tumour biopsy 15.0, Karnofsky Performance Status 23.0% 77.0%, MSKCC score poor 23.0%

Answer: We thank very much the reviewer for his comments on presentation style and will take them into account in the revised manuscript.

We sincerely appreciate the time and effort that you dedicated to providing feedback on our manuscript and we are grateful for the insightful comments on and valuable improvements to our paper.

Reviewer 2 Report

Dear Authors, I have read with interest your manuscript. The paper addresses an interestingissue regarding renal cell carcinoma, given the fact that it is part of  frequently diagnosed neoplasms, with frequent relapses and high mortality.

I would like to address a few suggestions/ questions:

1.  Can the type of surgical intervention, namely partial or total nephrectomy or only biopsy of the tumor, be associated with the evolution of patients towards second line therapy? Is the type of intervention of prognostic value?

2.   In the study, the patients were divided into 2 categories nlr<3 and nlr>3; considering that values ​​below 3 are not considered pathological and that values ​​between 2-3 can also exist in inflammation, atherosclerosis, is an nlr<3 unfavorable prognostic factor?

3.  Did the patients with nlr>3 have active infections at the time of inclusion in the study?

I think it is a very interesting topic to discuss, because given the burden of neoplasm worldwide, we need more accurate prognostic tools, to a better management for his patients. I congratulate you on this paper.

Author Response

We are grateful to the reviewer for the careful evaluation of our manuscript and for the constructive suggestions; all of them helped us to further improve the quality of our manuscript. We appreciate the thorough work of the reviewer and all the comments. We believe that we could fix his suggestions.

Thank you very much for your encouraging comments: “I think it is a very interesting topic to discuss, because, given the burden of neoplasm worldwide, we need more accurate prognostic tools, to a better management for his patients. I congratulate you on this paper.”

Comment 1. Can the type of surgical intervention, namely partial or total nephrectomy or only biopsy of the tumor, be associated with the evolution of patients towards second-line therapy? Is the type of intervention of prognostic value?

Answer: As the reviewer kindly suggested, we extensively worked the prognostic role of surgery in metastatic kidney cancer.

Almost 90% of patients had undergone initial surgical resection, including radical nephrectomy (24,3%) or without lymph node dissection (55,4%) and partial nephrectomy (9,5%). Minimally invasive biopsy procedures were performed in 10% of cases. The overall survival did not differ significantly between patients treated with radical nephrectomy (21,73±18,24 months) and partial nephrectomy (22,43±23,25 months) but was reduced in patients with a diagnostic biopsy (8,88±5,98 months). Unfortunately, we found no statistically significant correlation between the surgical approach and clinical benefit in first and second-line therapy. We also add new tables and figures to support the results.

Comment 2.   In the study, the patients were divided into 2 categories nlr<3 and nlr>3; considering that values ​​below 3 are not considered pathological and that values ​​between 2-3 can also exist in inflammation, atherosclerosis, is an nlr<3 unfavorable prognostic factor?

Answer: We have answered what the reviewer so nicely requested in the Discussion section:

Accumulating evidence has established the role of NLR as a biomarker of increased immune activation in different disorders, including infections, autoimmune diseases, cardiovascular diseases, and metabolic syndromes. Even if most studies have explored the prognostic value of NLR in cancer patients with an optimal cut-off value above 3, a grey zone of NLR values between two and three may serve as an early warning of endothelial dysfunction, chronic vascular inflammation, and atherosclerosis. The chronic vascular inflammatory process plays a role in the pathogenesis of atherosclerosis, hypertension, diabetes, and obesity.

Prior evidence has shown that NLR is a reliable biomarker to predict cardiovascular risk. In patients with coronary artery disease, an NLR  over 2.13  independently predicted myocardial damage. Furthermore, NLR has also been investigated as a marker of disease activity and a predictor of relapse in autoimmune diseases. For example, D'Amico et al. demonstrated that an NLR > 1 in patients with multiple sclerosis strongly predicted disease activity and aggressive evolution.

In conclusion, even NLR<3  may be an unfavorable prognostic factor for RCC patients with underlying comorbidities.

Comment 3. Did the patients with nlr>3 have active infections at the time of inclusion in the study?

Answer: As the reviewer kindly suggested, we included the information above to support patients selected for the study: Exclusion criteria include short-term follow-up ( <6 months), active autoimmune disease, evidence of active infection before initiating any systemic therapy, second primary cancers, brain metastases, and non-clear cell histology.

We sincerely appreciate all your valuable comments and suggestions, which helped us in improving the quality of the manuscript.

Author Response

We are grateful to the reviewer for the careful evaluation of our manuscript and for the constructive suggestions; all of them helped us to further improve the quality of our manuscript. 

Round 2

Reviewer 1 Report

The article has made significant improvements after modification, but the two issues raised earlier have not been fully revised. I would like to raise them again and hope to make careful revisions.

 In the articles and tables, there are a large number of “ , ” should be replaced with "."

The accuracy of Table 1 should be unified, Line  Tumour biopsy 15.0, Karnofsky Performance Status 23.0% 77.0%, MSKCC score poor 23.0%

Author Response

 We thank the reviewer again for the close reading of our article and all the constructive and detailed comments.

Reviewer 3 Report

The authors have satisfied all the requests, the work can be accepted in the present form.

Just make a little correction in figure 1, "prior" instead of "piror"

Author Response

We sincerely thank the reviewer again for providing constructive feedback to improve our manuscript!
